# Sequoia: A Software Framework to Unify Continual Learning Research

## Abstract

The field of Continual Learning (CL) seeks to develop algorithms that accumulate knowledge and skills over time through interaction with non-stationary environments. In practice, a plethora of evaluation procedures (*settings*) and algorithmic solutions (*methods*) exist, each with their own potentially disjoint set of assumptions. This variety makes measuring progress in CL difficult. We propose a taxonomy of settings, where each setting is described as a set of *assumptions*. A tree-shaped hierarchy emerges from this view, where more general settings become the parents of those with more restrictive assumptions. This makes it possible to use inheritance to share and reuse research, as developing a method for a given setting also makes it directly applicable onto any of its children. We instantiate this idea as a publicly available software framework called *Sequoia*, which features a wide variety of settings from both the Continual Supervised Learning (CSL) and Continual Reinforcement Learning (CRL) domains. Sequoia also includes a growing suite of methods which are easy to extend and customize, in addition to more specialized methods from external libraries. We hope that this new paradigm and its first implementation can help unify and accelerate research in CL. You can help us grow the tree by visiting (this GitHub URL).

## 1 Introduction

With the growing interest in developing methods robust to changes in the data distribution, research in continual learning (CL) has gained traction in recent years (Delange et al., 2021; Caccia et al., 2020; Parisi et al., 2019; Lesort et al., 2020). CL enables models to acquire knowledge from non-stationary data, learning new and possibly more complex tasks, while retaining performance on previously-learned tasks.

To instantiate a CL problem, one must first make assumptions about the data distribution and set constraints to enforce non-stationary learning. For instance, assumptions are often made about the type and number of tasks, the task boundaries, or the availability of task labels, while constraints often relate to memory, compute, or time allowed to learn a task. Combinations of assumptions, rules, and datasets have resulted in a multitude of settings (Khetarpal et al., 2020).

We argue that the increased popularity of CL, combined with a lack of unification — in part due to the large number of settings and the absence of well-defined applications — has led to a "research jungle" that may be slowing down progress.

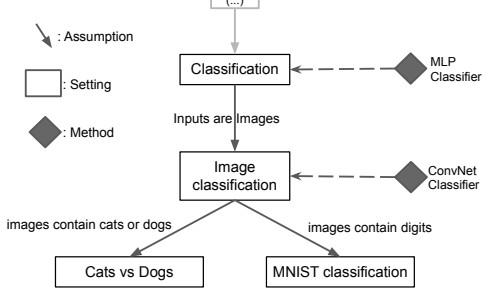

Figure 1: Example of a simple tree of settings that shows the core principle of our framework: research settings can be organized into a hierarchy based on their assumptions. Methods have a set of basic assumptions which correspond to their *target* setting (dashed arrows), and can be applied onto any of their descendants. In this trivial example, both MLP and ConvNet classifiers are applicable to the MNIST classification setting and their performances are directly comparable, even though they were created for slightly different settings. Sequoia applies this principle to the field of Continual Learning.

We identify some of the main challenges associated with the lack of unification in CL:

**i) Evaluation.** Methods in CL are often studied under a small subset of the available settings, making it difficult to evaluate them, as their problem domains don't always overlap. Consequently, it is challenging to determine if a method will generalize beyond the setting it was designed for. To add to this, continual reinforcement learning (CRL) poses further challenges in evaluation due to the lack of a clear distinction between training and testing phases (Khetarpal et al., 2018). Moreover, resource consumption is a critical factor for evaluation of CL methods which is often overlooked due to the lack of standardized evaluation protocols. Thus, there is a need for standardization of the infrastructure used to evaluate CL methods.

**ii) Reproducibility.** In order to analyze specific properties of novel methods, researchers tend to re-implement baselines and adapt them to their particular needs (Henderson et al., 2018). These baselines are often not described in enough detail to ensure reproducibility, e.g. a prescribed hyper-parameter search strategy, computational requirements, open source libraries, etc.

**iii) CSL and CRL evolve in silos.** Continual supervised learning (CSL) and continual reinforcement learning (CRL) are considered to be independent settings in the literature and thus, they are evolving separately. However, most methods in one field can be instantiated in the other, resulting in duplicate efforts such as replay for CSL (Rebuffi et al., 2017; Lesort et al., 2019a; Shin et al., 2017; Lesort et al., 2019b; Prabhu et al., 2020) and replay for CRL (Traoré et al., 2019; Rolnick et al., 2019; Kaplanis et al., 2020). To this end, we advocate that the unification of both fields would greatly reduce these duplicate efforts and accelerate CL research.

In this work, we present Sequoia, a unifying software framework for CL research, as a solution for jointly addressing these issues. We describe how *settings* differ from one another in terms of their *assumptions* (e.g., are task IDs observed or not). This perspective gives rise to a hierarchical organization of CL settings, through which methods become directly reusable by inheritance, thus greatly reducing the amount of work involved in developing and evaluating methods in CL.

**Key Contributions** of this work are 1) a general unified framework that systematically organizes CL settings; 2) *Sequoia*, a software framework that can serve as a universal platform for CL research.

## 2    A Unifying Framework for CL Research

To construct our unifying framework, we first represent each CL setting as a set of shared assumptions. More general settings make fewer assumptions and vice versa. Settings can then be organized in a hierarchy where adding/removing an assumption yields a child/parent setting (Fig. 2).

We formalize the framework using a hidden-mode Markov decision process (HM-MDP), a special case of a POMDP (Choi et al., 2000). HM-MDPs comprise an observation space $\mathcal{X}$, an action space $\mathcal{A}$, a context space $\mathcal{Z}$ (we also refer to contexts as tasks), and a feedback function $r$. Here the full state space is a concatenation of the observation space and task space $\mathcal{S} = \mathcal{X} \times \mathcal{Z}$. As such, the hidden context variable $z \in \mathcal{Z}$ defines the dynamics of the environment $p(x'|x, a, z)$ for observations $x', x \in \mathcal{X}$ and action $a \in \mathcal{A}$. The feedback function $r(x, a, z)$ provides an agent $\pi(a|x)$ (e.g. a supervised model) the value of performing action $a$ after receiving observation $x$. The context variable follows a Markov chain $p(z'|z)$. The dynamic context enables modelling CL task/context-change. A change in the context variable is called a *task boundary*.

In the next section (§ 2.1) we show that by restricting the different elements of the HM-MDP we recover different CL settings. Then in § 2.2 we discuss differences between continual supervised (CSL) and continual reinforcement learning (CRL). We end in § 2.3, by presenting additional assumptions that are relevant to CL problems.

### 2.1    Continual Learning Assumptions

Assumptions related to CL can be arranged into a hierarchy, as illustrated in the the central portion of Fig. 2. These settings cover most, if not all the current CL literature. We start from the most general setting: continuous task-agnostic CL (Zeno et al., 2018) and add assumptions one by one.

**Continuous Task-Agnostic CL** is our most general setting. The context variable is continuous $\ddagger \in \mathbb{R}$. This setting allows for different kinds of drifts in the environment, including smooth task

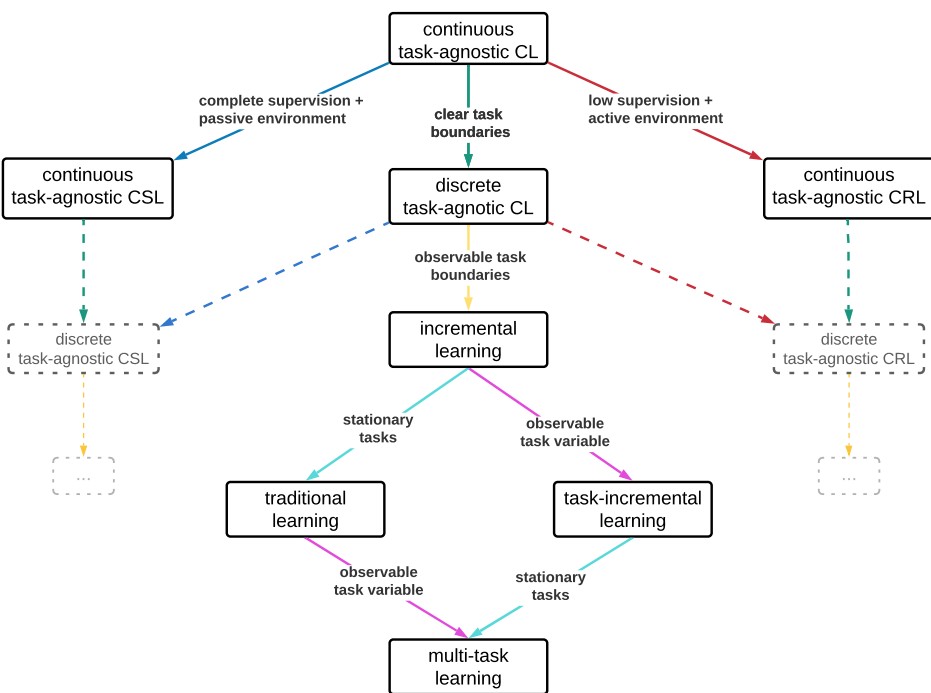

Figure 2: **Sequoia - The Continual Learning Research Tree.** Continual learning research settings can be organized into a tree, in which more general settings (parents) are linked with more restricted settings (children) by the differences in assumptions between them. Settings generally become more challenging the higher they are in this hierarchy, as less information becomes available to the method. The central portion of the tree shows the assumptions specific to CL, while the highest lateral branches indicate the choice of either supervised or reinforcement learning, which we consider to be orthogonal to CL. By combining either with the central assumptions, settings from Continual SL and Continual RL can be recovered to the left and right, respectively.

boundaries, i.e. slow drift (Zeno et al., 2018). This setting is *task-agnostic*, meaning that the context variable $z$ is unobserved. Because the context is allowed to drift slowly, it can be more challenging for the methods to infer when a task has changed enough to compartmentalize the recently acquired knowledge before adapting to the new task. In RL, this setting is analogous to the DP-MDP (Xie et al., 2020; Chandak et al., 2020b). In SL, it has also been studied in e.g. Zeno et al. (2018); Aljundi et al. (2019a;b).

**Discrete Task-Agnostic CL** assumes clear (or well-defined) task boundaries and so a discrete context variable $\mathcal{Z} \in \mathbb{N}$. In this setting the context can shift in a drastic way, still without the agent being explicitly noticed. Some cases where this setting has been studied are Choi et al. (2000); Riemer et al. (2018) for RL and Caccia et al. (2020); He et al. (2019); Harrison et al. (2019) for SL.

**Incremental Learning** (IL) relaxes the task-agnostic assumption: the task boundaries are observable. This is akin to augmenting the observation with a binary variable that is set to 1 when $z' \neq z$ and 0 otherwise. In doing so, the algorithm does not need to perform *task-boundary detection*. In SL, some well-known IL settings include class-IL and domain-IL distinguished by their *disjoint action space* and *shared action space*, respectively. This is discussed in § 2.3.

At this point in the CL hierarchy, the tree branches in two directions, depending on the order of remaining assumptions (see Fig. 2). We will first explain the right sub-tree.

**Task-Incremental Learning** (task-IL) assumes a fully-observable context variable available to the agent $\pi(a|x, z)$. In the literature, observing $z$ is analogous to knowing the *task ID* or *task label*. In this simpler CL setting, forgetting can be prevented by freezing a model at the completion of each task and using the task-ID to retrieve it for evaluation.

The following settings remove the non-stationarity assumption in the contexts/tasks and are often used to set an upper-bound performance for CL methods.

**Multi-task Learning** removes the non-stationarity in the environment dynamics and the feedback function as it assumes a stationary context variable $p(\boldsymbol{z}'|\boldsymbol{z}) = p(\boldsymbol{z}')$. When the contexts are stationary, there is no *catastrophic forgetting* (CF) (French, 1999) problem to solve. Multi-task learning assumes a fully-observable task variable.

**Traditional Learning** branches off incremental CL and assumes a stationary environment. It is the vanilla supervised setting machine learning defaults to. In our framework, it can be seen as a multi-task learning problem where the task variable isn't observable. However, a more natural view of this setting is to simply assume a single task/context.

## 2.2 SUPERVISED LEARNING AND REINFORCEMENT LEARNING ASSUMPTIONS

So far we have introduced settings and assumptions that revolve mainly around the type and presence of non-stationarity in the environment and the information observed by the agent. These have allowed us to define the CL problem. To bring all of CL research under one umbrella, we introduce two assumptions, orthogonal to the previous ones, to recover RL and SL settings. With these assumptions, methods for a given CL setting are applicable to both its CSL and CRL versions, as in Kirkpatrick et al. (2017); Fernando et al. (2017).

Below we use the term observation as a *state* in RL parlance and the actions as *predictions* in SL parlance. Also, we assume a single context or task.

**Level of feedback:** In RL, the feedback function $r(\boldsymbol{x}, \boldsymbol{a}, \boldsymbol{z})$ returns a *reward* that informs the agent about the value of performing action $\boldsymbol{a}$ after receiving observation $\boldsymbol{x}$ in context $\boldsymbol{z}$. In SL however, the feedback function is generally both directly known by the agent and differentiable, which allows the agent to simultaneously consider the value of all actions for a particular observation. This feedback is computed based on a *label* when the action space is discrete (classification) or a *target* when it is continuous (regression). The feedback level is a key differentiating feature between RL and SL.

**Active vs passive environments:** In RL, it is generally assumed that the agent's action has an effect on the next observation or state.[1] In other words, the dynamics of the environment $p(\boldsymbol{x}'|\boldsymbol{x}, \boldsymbol{a}, \boldsymbol{z})$ are action-dependant and we call this an *active* environment. In SL the agent is generally assumed to not influence the next observation i.e. $p(\boldsymbol{x}'|\boldsymbol{x}, \boldsymbol{a}, \boldsymbol{z}) = p(\boldsymbol{x}'|\boldsymbol{x}, \boldsymbol{z})$. The environment is thus referred to as being *passive* in these cases.

As seen in Figure 2, the two aforementioned assumptions are combined into a single assumption for SL (blue, left) and for RL (red, right). By combining either the RL or SL assumption along with those from the the central CL "trunk", settings from CSL and CRL are recovered. Future versions of Sequoia will decouple these assumptions to enable settings such as bandits and imitation learning.

## 2.3 ADDITIONAL ASSUMPTIONS

Additional assumptions can be added on top of the ones described above to recover additional research settings. For example, a useful assumption in CL experiments is the one of disjoint versus joint action space, i.e. whether the contexts/tasks share a same action space, or whether that space is different for each task. In CSL, this assumption differentiates *class-incremental learning* from *domain-incremental learning* (van de Ven & Tolias, 2019). In Farquhar & Gal (2018), where it is referred as the *shared output space* assumption, a disjoint action space greatly increases the difficulty of a setting in terms of forgetting. In CRL however, the studied settings mostly have a joint action space, with the notable exception in the work of Chandak et al. (2020a).

Other assumptions could also be relevant in defining a continual learning problem. For instance, the action space being either discrete or continuous, resulting in classification and regression CSL problems, respectively; a particular structure being required of the method's actions, as in image segmentation problems; an episodic vs non-episodic setting in RL; context-dependant (Caccia et al., 2020) versus context-independent feedback functions; and many more.

---

[1]The *bandit* setting is one notable exception to this rule.

## 3   SEQUOIA - A SOFTWARE FRAMEWORK

Alongside this unifying perspective, we introduce *Sequoia*, an open-source python framework. Each setting described above is instantiated as a class in a tree-shape inheritance hierarchy. Sequoia is designed to address some of the issues associated with Continual Learning research, previously described in § 1.

First, we establish a clear **separation of concerns** between research problems and the solutions to such problems. We establish this separation through two core abstractions: `Setting` and `Method`. This decoupling greatly helps to evaluate methods, since the logic for each component is cleanly separated, and extracting a component and reusing it elsewhere becomes possible. An example of a Method is shown in Listing 2.

Second, to help bridge the gap between the CRL and CSL domains, Sequoia uses `Environment` as the interface between methods and settings. `Environment`s extend the familiar abstractions from OpenAI `gym`, to also include supervised learning datasets, making it possible to develop methods that are applicable in both the CRL and CSL domains. `Environment` will be described in § 3.2.

Finally, Sequoia uses inheritance to **make methods directly reusable across settings**. By organizing research settings into a tree-shaped inheritance hierarchy, along with their environments, observations, actions, and rewards, Sequoia enables methods developed for any particular setting to be applicable onto any of their descendants, since all the objects the method will interact with will inherit from those they were designed to handle. This mechanism has the potential to greatly improve code reuse and reproducibility in CL research.

This section first describes each of these abstractions in more detail, after which the currently available settings and methods are described. § 4 will then provide a demonstration of the kind of large-scale empirical studies which are made possible through the use of this new framework.

```python
from sequoia.settings.sl import *
from sequoia.settings.rl import *
from sequoia.methods import BaseMethod

method = BaseMethod(learning_rate=1e-3)

for setting in [
    ContinuousTaskAgnosticSLSetting("mnist"),
    ContinuousTaskAgnosticRLSetting("cartpole"),
    DiscreteTaskAgnosticSLSetting("mnist"),
    DiscreteTaskAgnosticRLSetting("cartpole"),
    IncrementalSLSetting("mnist"),
    IncrementalRLSetting("cartpole"),
    TaskIncrementalSLSetting("mnist"),
    TaskIncrementalRLSetting("cartpole"),
    MultiTaskSLSetting("mnist"),
    MultiTaskRLSetting("cartpole"),
    TraditionalSLSetting("mnist"),
    TraditionalRLSetting("cartpole"),
]:
    results = setting.apply(method)
    results.summary()
    results.make_plots()
```

Listing 1: Code snippet, using Sequoia to evaluate a method in multiple settings. This particular example is made possible by the `BaseMethod`, which is applicable to all settings.

**Relation with other frameworks:** Sequoia is in no way competing with existing tools and libraries which provide standardized benchmarks, models, or algorithm implementations. On the contrary, Sequoia benefits from the development of such frameworks.

In the case of libraries that introduce standardized benchmarks, they can be used to enrich existing settings with additional datasets or environments (Douillard & Lesort, 2021; Brockman et al., 2016), or even to create entirely new settings. Likewise, frameworks which introduce new models or algorithms (Lomonaco et al., 2021; Raffin et al., 2019; Wolczyk et al., 2021) can also be used to create new Methods or to add new backbones to existing Methods within Sequoia. External repositories can register their own methods through a simple plugin system. The end goal for Sequoia is to provide the research community with a centralized catalog of the different research frameworks and their associated methods, settings, environments, etc. The following sections will show examples of such extensions.

### 3.1   SETTINGS

A `Setting` can be viewed as a configurable evaluation procedure for a `Method`. It creates various training environments, evaluates the method, and finally returns some `Results`. These results contain various metrics relevant to the setting. The training/testing routine for each setting is imple-

mented according to the evaluation protocol of that setting in the literature. An example of applying a method onto multiple settings is shown in Listing 1.

Concretely, settings create the training, validation, and testing environments that a method interacts with. This interface also makes it possible for methods to leverage PyTorch-Lightning (Falcon et al., 2019) to perform high-performance training of their models.[2] For more information on the interactions between Sequoia and PyTorch-Lightning, see App. C.

Settings are available for each combination of the CL assumptions, along with the choice of one of RL / SL (as illustrated in Figure 2), for a total of 12 settings.[3] These two "branches" (one for CRL and the other CSL) form the basis of Sequoia's eponymous tree of settings. Each setting inherits from one or more parent settings, following the above-mentioned organization.

Every `Setting` is created by extending a more general `Setting` and adding additional assumptions. This inheritance relationship from one setting to the next also extends to the setting's environments (`Environment`), as well as the objects (`Observations`, `Actions`, and `Rewards`) they create. See App. D for an illustration of this principle.

## 3.2 ENVIRONMENTS

Settings in Sequoia create training, validation, and testing `Environment`s, which adhere to both the `gym.Env` and the `torch.DataLoader` APIs. This makes it easy for SL researchers to transition to RL and vice-versa. These environments receive `Actions` and return `Observations` and `Rewards`. `Observations` contain the input samples $x$, and may also contain task labels for each sample, depending on the setting. These objects have the same structure in both RL and SL settings. However, as described in § 2.2, in SL, `Actions` correspond to the predictions, while `Rewards` correspond to targets or labels. These objects are defined by the `Setting` and follow the same pattern of inheritance as the settings themselves. The structure of these objects are reflected in the environment's observation, action, and reward spaces, which are used within methods to create their models.

**Supervised Learning environments** Sequoia supports most of the datasets traditionally used in continual supervised learning research, through its use of the *Continuum* package (Douillard & Lesort, 2021). The list of supported datasets is available in Table 1.

**Reinforcement Learning environments** Through its close integration with `gym`, Sequoia is able to use any gym-compatible environment as the "dataset" used by its RL settings. For settings with multiple tasks, Sequoia simply needs a way to sample new tasks for the chosen environment. This mechanism makes it easy to add support for new or existing gym environments. An example of this is included in App. D.

Sequoia creates continuous or discrete tasks, depending on the choice of setting and dataset/environment. For example, when using one of the classic-control environments from gym such as `CartPole`, tasks are created by sampling a new set of values for the environment constants such as the gravity, the length of the pole, the mass of the cart, etc. This is also the case for the well-known `HalfCheetah`, `Walker2d`, and `Hopper` MuJoCo environments, where tasks can be created by introducing changes in the environmental constants such as gravity. Continuous tasks can thus easily be created in this case, as the environment is able to respond dynamically to changes in these values at every step, and the task can evolve smoothly by interpolating between different target values.

Other gym environments become available when using RL settings with discrete tasks (i.e. all settings that inherit from `DiscreteTaskAgnosticRLSetting`), as it becomes possible to simply give Sequoia a list of environments to use for each task, and the constructed Setting will then use them as part of its evaluation procedure.

We use this feature to construct continual variants of the MT10, MT50 benchmarks from Meta-World (Yu et al., 2019), as well to replicate the CW10 and CW20 benchmarks introduced in Wolczyk

---

[2]It is important to note that methods are in no way required to use PyTorch-Lightning.

[3]Other common SL settings, such as Domain-Incremental learning are also available in Sequoia, but they rely on an additional family of assumption, and are thus omitted from the main portion of this paper for sake of brevity and clarity.

| Methods | SL | `BaseMethod`.{base, EWC, PackNet }, PNN, replay, HAT, CN-DPM `Avalanche`.{naive, AGEM, CWR*, EWC, Gdumb, GEM, LWF, replay, SI} |
| | RL | `BaseMethod`.{base, EWC, PackNet }, PNN `stable_baselines3`.{A2C, DDPG, DQN, PPO, SAC, TD3} `continual_world`.{SAC, AGEM, EWC, VCL, PackNet, L2 reg., MAS, replay} |
| Environments | SL | `continuum`.{{K,E,Q,Fashion}MNIST, Cifar10(0), ImageNet100(0), Core50, Synbols} |
| | RL | `gym`.{Hopper, Half-Chettah, Walker2d, CartPole, Pendulum, MontainCar} Monsterkong, `metaworld`.{MT10, MT50}, `continual_world`.{CW10, CW20} |
| Metrics | | {Transfer Matrix, forward transfer, backward transfer, Average final performance, Online Training Performance} × |
| | SL | {loss, accuracy} |
| | RL | {loss, total reward, average reward, episode length} |

Table 1: **Sequoia's methods, environments and metrics.** Most of the RL settings in Sequoia can be passed custom environments to use for each task. This makes it possible to use virtually any gym environment to create custom incremental RL settings. The environments listed here are those explicitly supported in Sequoia, where multiple tasks can be sampled within a single environment.

```python
import gym
from sequoia.settings import Setting, Environment, Observations, Actions
from sequoia.methods import Method

class DemoModel:
    def forward(self, observations: Observations) -> Actions:
        ...

class DemoMethod(Method, target_setting=Setting):
    """ Pseudocode for a Method that targets a given Setting. """
    def configure(self, setting: Setting):
        # Called by the setting before training begins.
        self.model = DemoModel(setting.observation_space, setting.action_space,
                               setting.reward_space, nb_tasks=setting.nb_tasks)
        self.optimizer = Adam(...)

    def fit(self, train_env: Environment, valid_env: Environment):
        # Train a model using these environments from the setting.
        # Note: all Environments are gym environments. More on this later.
        for epoch in range(self.n_epochs):
            self.model.train_epoch(train_env)
            self.model.validation_epoch(valid_env)

    def get_actions(self, observations: Observations) -> Actions:
        # Called by the setting for inference (at test-time).
        actions = self.model(observations)
        return actions

    def on_task_switch(self, task_id: Optional[int]):
        # Gets called on task boundaries, depending on the setting.
        self.model.prepare_for_new_task(new_task=task_id)
```

Listing 2: Pseudocode for creating a new Method.

et al. (2021). Sequoia also introduces a non-stationary version of the *MonsterKong* environment, described in App. E.2. A more complete list of the supported environments is shown in Table 1.

### 3.3 METHODS

Methods hold the logic related to the model and training algorithm. When defined, each method selects a "target setting" from those available in the tree. A method can be applied to its target setting as well as any setting which inherits from it (i.e. any setting which is a child node of the target setting). We now provide a brief description of the different types of Methods available in Sequoia. An illustration of the Method API can be seen in Listing 2.

**General methods:** Methods in Sequoia can target settings from either the RL or SL branches of the tree. Additionally, it is also possible to select one of the settings from the central CL branch - for instance, Incremental Learning. This makes methods applicable to both the CRL and CSL variants of that setting. One such method is the `BaseMethod`, which can be applied to any setting in the tree,

and is provided as a modular, customizable, jumping off point for new users. This `BaseMethod` is equipped with modules for task inference and multi-head prediction. See App. B for a more in-depth discussion of its features and capabilities.

**Supervised Learning Methods:** Sequoia benefits from other CL frameworks such as Avalanche (Lomonaco et al., 2021). Avalanche offers both standardized benchmarks as well as a growing set of CL methods, which are referred to as *strategies* in Avalanche. Sequoia reuses these strategies as `Method` classes. See Table 1 for a complete list of such methods.

**Reinforcement Learning Methods:** Settings in Sequoia produce Environments, which adhere exactly to the Gym API. It is therefore easy to import existing RL tools and libraries and use them to create new methods.

As an example, here we enlist the help of a specialized framework for RL, namely stable-baselines3 (Raffin et al., 2019) . The A2C, PPO, DQN, DDPG, TD3 and SAC algorithms from SB3 were easily introduced as new `Method` classes in Sequoia, without duplicating any code. These methods are applicable onto any of the RL settings in the tree.

In addition to these RL backbones from SB3, Continual reinforcement learning methods are also available. These methods were adapted from the work of Wolczyk et al. (2021), which introduced the CW10 and CW20 benchmarks for continual learning, based on sequences of tasks from Meta-World (Yu et al., 2019). The authors also provided implementations for CRL algorithms, built on top of a SAC(Haarnoja et al., 2018) backbone. These algorithms were adapted from their original implementation and made available as CRL methods in Sequoia (see Table 1 for full the list).[4]

## 4 EXPERIMENTS

Sequoia's design makes it easy to conduct large-scale experiments to compare the performance of different methods on a given setting, or to evaluate the performance of a given method across multiple settings and datasets. We illustrate this by performing large-scale empirical studies involving all the settings and methods available in Sequoia, both in CRL and CSL. Each study involves up to 20 hyper-parameter configurations for each combination of setting, method, and dataset, in both RL and SL, for a combined total of $\approx 8000$ individual completed runs. The rest of this section provides a brief overview of these experiments, which are also publicly available at https://wandb.ai/sequoia/.[5] All results are reproduced in App. F in a larger format and accompanied with further analysis.

**Continual Supervised Learning.** As part of the CSL study, we use some of the "standard" image classification datasets such as MNIST (LeCun & Cortes, 2010), Cifar10, and Cifar100 (Krizhevsky et al., 2009). Furthermore, we also include the Synbols (Lacoste et al., 2020) dataset, a character dataset composed of two independent labels: the characters and the fonts. (see App. E.1 for the motivation). Exhaustive results can be found at https://wandb.ai/sequoia/csl_study.

A sample of these results is illustrated in Figure 3, which shows results of various methods in the class-IL and task-IL settings in terms of their final performance and runtime. We note that some `Avalanche` methods achieve lower than chance accuracy in task-IL because they do not use the task label to mask out the classes that lie outside the tested task.

**Continual Reinforcement Learning.** We apply the RL methods from SB3 on multiple benchmarks built on HalfCheetah-v2, Hopper-v2, MountainCar-v0, CartPole-v0, MetaWorld-v2 (details in App. E). We also introduce a new discrete domain benchmark, namely Continual-MonsterKong, that we developed to study forward transfer in a more meaningful way (see App. E.2 for more details). Complete results are available at https://wandb.ai/sequoia/crl_study.

A sample of these results is in Figure 4. It presents various methods in the traditional and incremental learning settings with their final performance, online performance and normalized runtime.

In Table 2, we apply the `continual-world` methods, built on top of SAC, on a incremental RL benchmark inspired by Mendez et al. (2020) (see App. E.3 for more details). Finally, Figure 11 shows the transfer matrix achieved by one such algorithm, namely PPO (Schulman et al., 2017).

---

[4]While most other methods use PyTorch these methods are implemented using Tensorflow.
[5]We will update these sample studies periodically to reflect all future improvements made to the framework.

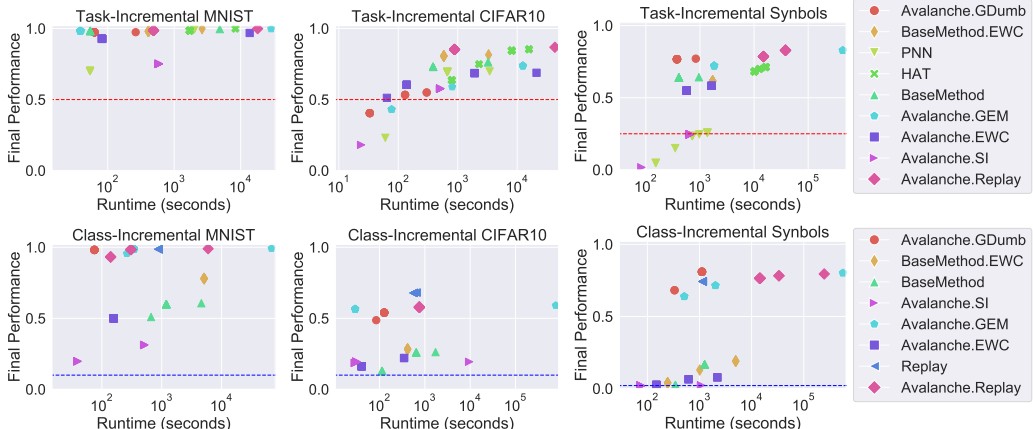

Figure 3: **Incremental Supervised Learning results**. Final performance (vertical axis) is plotted against runtime (horizontal axis). The methods achieving the best trade-off lie closer to the top-left of the figures. Task-Incremental and Class-Incremental results are presented on the top and bottom row, respectively. The dotted line shows chance accuracy for each setting-dataset combination. For each methods, several trials are presented depending on metrics composed of linear combination of final performance and (normalized) runtime. GEM and GDumb achieve the best tradoff, although the latter cannot make predictions in an online manner and thus serves more as a reference point.

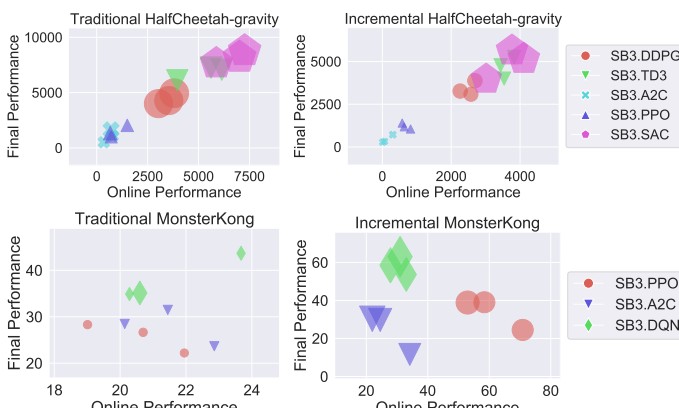

|  | FP (↑) | OP (↑) | R (↓) |
|---|---|---|---|
| Fine-tuning | 309 | 278 | **17.1** |
| L2 | 473 | 236 | 19.2 |
| EWC | 529 | 230 | 20.4 |
| MAS | 657 | 238 | 20.0 |
| PackNet | 1002 | 267 | 20 |
| Perfect Memory | **1134** | 271 | 20.2 |
| A-GEM | 776 | **281** | 24.5 |

Table 2: **Incremental RL results.** Multiple CRL methods, all built on top of SAC, are tested on the Hopper-Bodyparts benchmarks. FP and OP stands for final and online performance, respectively, whereas R stands for runtime, reported in hours. Results are averaged over 5 seeds. All CRL methods outperformed the Fine-tuning baseline, validating their efficacy. Experience replay with a Perfect Memory achieves the best retained performance on all tasks, followed closely by PackNet.

Figure 4: **Impact of the RL backbone algorithm in Traditional and Incremental RL**. Final performance (vertical axis) is plotted against online performance (horizontal axis). The bubbles' size indicates the normalized runtime of the methods. The methods achieving the best trade-off lie closer to the top-right of the figures and have smaller bubble size. Datasets are presented in each row and settings are presented in each column. For each method, several trials are presented depending on metrics composed of linear combination of final performance and online performance.

## 5 CONCLUSION

In this work, we introduce Sequoia: a publicly available framework to organize virtually all research settings from both the fields of Continual Supervised and Continual Reinforcement learning. Sequoia also makes methods are directly reusable by contract across settings using inheritance. It is our hope that Sequoia will be useful to new and experienced researchers in CL. Further, the principles used to construct this framework for CL could very well be applied to other fields of research, effectively growing the tree towards new and interesting directions. We welcome suggestions and contributions to that effect in our GitHub page at (this GitHub url).

**Reproducibility statement**  To facilitate reproducing results of our experiments, we include an anonymized version of the Sequoia codebase. All results from the experiments of § 4 can be observed at https://wandb.ai/sequoia. Each run includes the exact command used, as well as the git state, the complete system specification, hyper-parameter configurations, and random seeds used.

While most sources of randomness are accounted for in Sequoia, we are still in the process of making settings entirely deterministic given a random seed. In other words, for some combinations of settings and methods, launching two runs with the exact same arguments and seeds do sometimes produce different results. Making settings and methods entirely deterministic is part of the plans for future work in this project.

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

## A    SUPPORTED METHODS

One of Sequoia's biggest strength is how easy it is to extend. Most methods in Sequoia are the result directly reusing existing implementations from other frameworks and repositories, such as AvalancheLomonaco et al. (2021), Stable-Baselines3Raffin et al. (2019) and Continual WorldWolczyk et al. (2021). Table 3 shows all the methods currently available in Sequoia.

| Method | Target setting |
|---|---|
| `BaseMethod` | `Setting` (all) |
| `BaseMethod`.EWC Kirkpatrick et al. (2017) | `Setting` (all) |
| `BaseMethod`.PackNet Mallya & Lazebnik (2018) | Incremental Learning (RL + SL) |
| replay | Incremental SL |
| CN-DPM Lee et al. (2020) | Continual SL |
| HAT Serrà et al. (2018) | Task-Incremental SL |
| PNN Rusu et al. (2016) | Incremental SL |
| `Avalanche`.naive Lomonaco et al. (2021) | Incremental SL |
| `Avalanche`.AGEM Chaudhry et al. (2019) | Incremental SL |
| `Avalanche`.cwr_star Lomonaco et al. (2021) | Incremental SL |
| `Avalanche`.EWC Kirkpatrick et al. (2017) | Incremental SL |
| `Avalanche`.Gdumb Prabhu et al. (2020) | Incremental SL |
| `Avalanche`.GEM Lopez-Paz & Ranzato (2017) | Incremental SL |
| `Avalanche`.LWF Li et al. (2019) | Incremental SL |
| `Avalanche`.replay | Incremental SL |
| `Avalanche`.SI Zenke et al. (2017) | Incremental SL |
| `stable-baselines3`.A2C Mnih et al. (2016) | Incremental RL |
| `stable-baselines3`.DDPG Lillicrap et al. (2015) | Continual RL |
| `stable-baselines3`.DQN Mnih et al. (2015) | Continual RL |
| `stable-baselines3`.PPO Schulman et al. (2017) | Continual RL |
| `stable-baselines3`.SAC Haarnoja et al. (2018) | Continual RL |
| `stable-baselines3`.TD3 Fujimoto et al. (2018) | Continual RL |
| `continual_world`.SAC Haarnoja et al. (2018) | Incremental RL |
| `continual_world`.AGEM Chaudhry et al. (2019) | Incremental RL |
| `continual_world`.EWC Kirkpatrick et al. (2017) | Incremental RL |
| `continual_world`.VCL Nguyen et al. (2018) | Incremental RL |
| `continual_world`.MAS Aljundi et al. (2018) | Incremental RL |
| `continual_world`.L2 regularization | Incremental RL |
| `continual_world`.PackNet Mallya & Lazebnik (2018) | Incremental RL |
| `continual_world`.Replay | Incremental RL |

Table 3: **Sequoia's methods support.** Each method specifies a target setting, listed on the right. Most methods are applicable in either RL or SL, while some can be applied to both. Methods also specify the "level of nonstationarity" they are prepared to handle, as a choice of one of Continual (which is also referred to as Continuous Task-Agnostic CL in § 2.1), Discrete, Incremental, Task-Incremental, Traditional, and Multi-Task.

## B    THE BASE METHOD

While developing a new `Method` in Sequoia, users are encouraged to separate the training logic from the networks used, the former being contained in the `Method`, and the latter in a model class, as advocated by PyTorch-Lightning (Falcon et al., 2019) (PL), a powerful research library, which we employ as part of this `BaseMethod`.

The `BaseMethod` is accompanied by the `BaseModel`, which acts as a modular and extendable model for CL Methods to use. This `BaseModel` adheres to PyTorch-Lightning's `LightningModule` interface, making it easy to extend and customize with additional callbacks and loggers. Likewise, the `BaseMethod` employs a `pl.Trainer`, which is able to train the `BaseModel` on the `Environment`s produced by any setting. Sequoia's Settings are also closely related to PL's

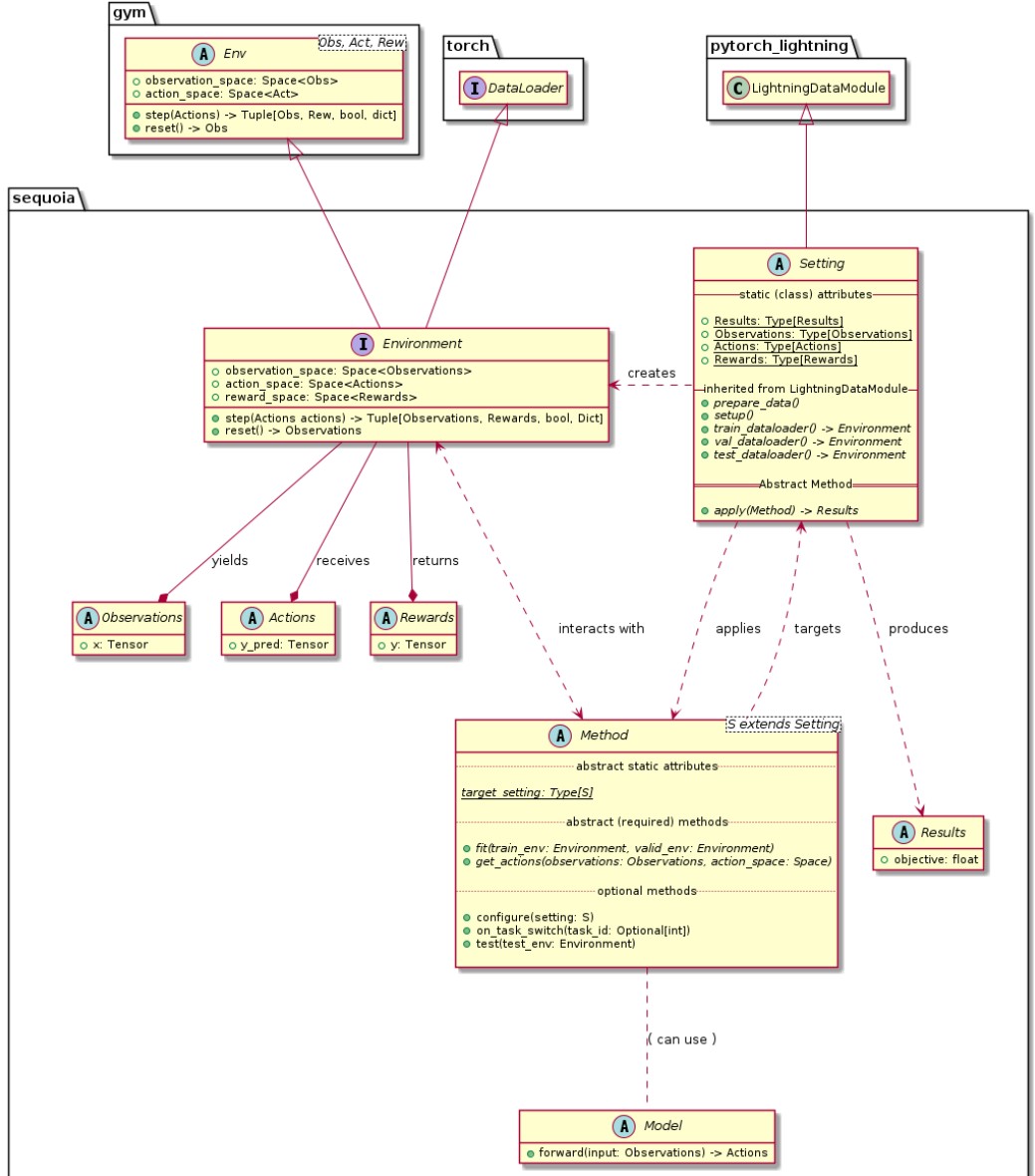

Figure 5: UML Diagram showing the main abstractions of Sequoia.

`DataModule` abstraction. See App. C for a further discussion of the relationship between Sequoia and Pytorch-Lightning.

Using this `BaseModel` when creating a new CL method can be particularly useful when transitioning from a CL `Setting` to its parent, as it comes equipped with most of the components required to handle such transitions (e.g. task inference, multi-head prediction, etc.) These components, as well as the underlying encoder, output head, loss function, etc. can easily be replaced or customized.

Additional losses can also be added to the `BaseModel` through a modular interface, which was explicitly designed to facilitate exploration of self-supervised learning research.

## C  ADDING NEW SETTINGS TO SEQUOIA

The very simple definition of the `Setting` abstract base class means that new Settings are not required to place themselves into our existing inheritance hierarchy, and could also not be related to continual learning at all! It is however preferable, whenever possible, to find the closest existing Setting within Sequoia and either add additional assumptions or remove extraneous ones, by creating the new Setting either below its closest relative or above it when adding or removing assumptions respectively.

The most general Setting in our current hierarchy - also referred to as the "root" setting - inherits from this abstract base class, while also building upon the elegant `DataModule` abstraction introduced by Pytorch-Lightning Falcon et al. (2019), in which the `DataModule` is the entity responsible for the preparation of the data, as well as for the creation of the training, validation and testing `DataLoaders`. Models in pytorch lightning can thus easily train and evaluate themselves through this standardized API, where dataloaders can be swapped out between experiments. Sequoia's main contributions can thus be viewed as taking this idea one step further, by 1) giving control of the "main loop" to this construct (through the addition of the `apply` method), 2) expanding this idea into the realm of Reinforcement Learning by moving from PyTorch's `DataLoaders` to a higher-level abstraction (`Environment`s), and 3) organizing these modules into an inheritance hierarchy.

Given how all current Sequoia Settings are instances of PL's `LightningDataModule` class, it is easy to use pytorch lightning for the training of Methods in Sequoia. This is one of the reasons why, for instance, the `BaseMethod` uses Pytorch Lightning's Trainer class in its implementation. However, the trainer-based API is not directly usable, due to the very nature of CL problems, in which there are training dataloaders for each task, which isn't currently possible through the standard Pytorch Lightning's API.

```python
from typing import List, Dict
from sequoia.settings.rl import make_continuous_task
from gym.envs.classic_control import CartPoleEnv
import numpy as np

# A Continuous task is a dict mapping from attributes to the values
# to be set on the environment:
ContinuousTask = Dict[str, float]

@make_continuous_task.register(CartPoleEnv)
def make_task_for_my_env(
    env: CartPoleEnv, step: int, change_steps: List[int], seed: int = None, **kwargs,
) -> ContinuousTask:
    # NOTE: task sampling should be reproducible given a `seed`.
    step_seed = seed * step if seed is not None else None
    rng = np.random.default_rng(step_seed)
    return {
        "gravity": 9.8 * rng.normal(1, 0.5),
        "masscart": 1.0 * rng.normal(1, 0.5),
        "masspole": 0.1 * rng.normal(1, 0.5),
        "length": 0.5 * rng.normal(1, 0.5),
        "force_mag": 10.0 * rng.normal(1, 0.5),
        "tau": 0.02 * rng.normal(1, 0.5),
    }
```

Listing 3: Example of how to add new RL environments to Sequoia. In this example, we register a function which will be used to sample continuous tasks for this environment, allowing it to become used as part of the Continuous Task-Agnostic Continual RL Setting and all of its descendants.

```python
import operator
from typing import List, Dict, Union, Callable
import numpy as np
import gym
from metaworld.envs.mujoco.sawyer_xyz.v2 import SawyerReachEnvV2
from metaworld import ML10
from sequoia.settings.rl.discrete import make_discrete_task

# In the case of Discrete RL settings, you can either return a
# 'continuous' task as before or a callable which will be
# applied onto the environment when a task boundary
# is reached:
ContinuousTask = Dict[str, float]
IncrementalTask = Union[ContinuousTask, Callable[[gym.Env], None]]

@make_discrete_task.register(SawyerReachEnvV2)
def make_discrete_task_for_metaworld_env(
    env: SawyerReachEnvV2,
    step: int,
    change_steps: List[int],
    seed: int = None,
    **kwargs,
) -> IncrementalTask:
    benchmark = ML10(seed=seed)
    rng = np.random.default_rng(seed)
    some_metaworld_task = rng.choice(benchmark.train_tasks)
    # NOTE: Equivalent to the following, but has the benefit of
    # being pickleable for vectorized envs:
    # return lambda env: env.set_task(some_metaworld_task)
    return operator.methodcaller("set_task", some_metaworld_task)
```

Listing 4: Example of how to add support for new RL environments to Sequoia in the case of discrete tasks, which are applied when a task boundary is reached. In this example, we register a function which will be used to sample discrete tasks for this environment, allowing it to become used as part of the Discrete Task-Agnostic RL setting and its descendants.

## D  ADDING NEW ENVIRONMENTS

New environments can be added to Sequoia by registering a new handler for creating new tasks, as can be seen in Listing 3 for continuous tasks, and in Listing 4 for discrete tasks.

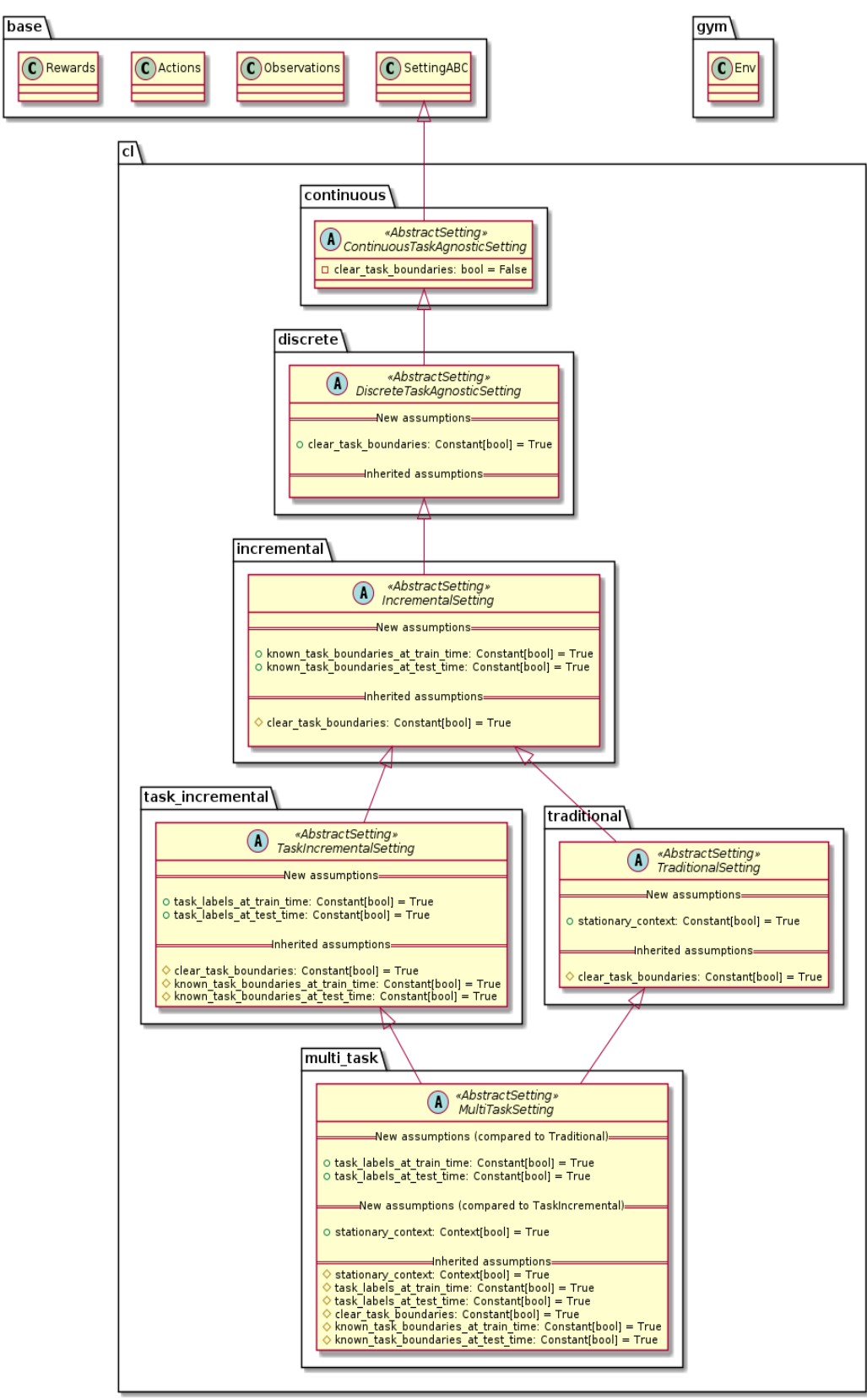

Figure 6: UML Diagram of the CL assumptions hierarchy. The CRL and CSL branches are not shown, but follow an identical structure.

# E BENCHMARK DETAILS

## E.1 SPLIT-SYNBOLS DATASET

Currently employed datasets can't be used sensibly to construct domain-incremental learning problems. Some have used MNIST to construct Permuted-MNIST and Rotated-MNIST, however, Farquhar & Gal (2018) have explained and demonstrated why such benchmarks are flawed and bias their results unfairly towards some methods. Motivated by this, we introduce Split-Synbols. Based on the Synbols dataset (Lacoste et al., 2020), a character classification dataset in which examples have an extra label corresponding to their font, one can easily construct sensible domain-incremental benchmark where e.g., a font would consist of a domain.

For the experiments, however, we opted for a class-incremental version to increase the difficulty. We prescribe a segmentation into 12 tasks to be learned sequentially, each consisting of a 4-way classification problem. Some example of Synbols character are displayed in Figure 7.

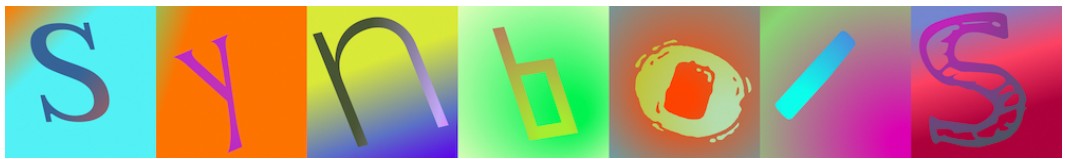

Figure 7: **Split-Synbols.** Example of Synbols character to classify.

## E.2 CONTINUAL-MONSTERKONG ENVIRONMENT

With rapid advancements in the field of deep RL, continual RL or *never-ending*-RL has witnessed rekindled interest towards the goal for broad-AI in recent years. While significant progress has been made in related domains such as transfer learning Taylor & Stone (2009), multi-task learning Ammar et al. (2014); Parisotto et al. (2016); Calandriello et al. (2014); Maurer et al. (2016); Barreto et al. (2019); Landolfi et al. (2019), and generalization in RL Farebrother et al. (2018), an outstanding bottleneck is the lack of standard tools to develop and evaluate CRL agents Khetarpal et al. (2020). A standardized benchmark will potentially enable rapid research and development of CRL agents. To this end, we propose a new CRL benchmark within the unified framework of Sequoia. In particular, we build the CRL benchmark leveraging the Pygame learning environment `MonsterKong` Tasfi (2016). MonsterKong is pixel-based, lightweight and has an easily-customizable domain, making it a good choice for evaluating continual learning agents.

Specifically, we design *tasks* through a variety of map configurations. These configurations vary in terms of the location of the goal and the location of coins within each level. We introduce randomness across runs of a task by varying the start locations of the agent. To incorporate the ability to evaluate across specific CRL characteristics, we leverage tasks to define CRL *experiments*. We design families of tasks leveraging the following abstract concepts: *jumping tasks* which require the agent to perform jumps across platforms of different lengths in order to collect coins and reach the goal, *climbing tasks* which require the agent to competently navigate ladders in order to collect coins and reach the goal, and tasks that combine both of these skills. The specific tasks leveraged as part of the CRL competition are depicted in Figure 8. The agent trains on each task for 200,000 steps.

**Experiment Details:** To evaluate the agents on the CRL benchmark, we follow the standard evaluation introduced above. Final performance reports accumulated reward per episode on all test environments, averaged over all tasks, after the end of training, whereas online performance is measured as the accumulated reward per episode on the training environment of the current task during training of all tasks. For the runtime score, we use set *max_runtime* of 12 hours and *min_runtime* to 1.5 hours. Lastly, the agents are allowed a maximum of 200,000 steps per task.

**Customization:** Ideally, CRL agents must be able to solve tasks by acquiring knowledge in the form of skills, be able to use previously acquired behaviors, and build even more complex behaviours over the course of its lifetime Ring (1997); Thrun & Mitchell (1995); Thrun & Schwartz (1995). While leveraging the MonsterKong environment, it is easy to introduce new environment layouts or modifications to existing layouts. Configurations could be customized to include arbitrary configurations

Figure 8: **Continual-MonsterKong.** We display the 8 tasks that constitute the benchmark in chronological order. The first two tasks test the agent's ability to jump between platforms, the second two test its ability to climb ladders and the last four combine both skills.

of coins, ladders, platforms, walls, monsters, fire balls, and spikes. Making custom environment elements is straightforward as well, so the environment can be modified to aligned with the properties of the CRL agent that we would like to test.

While in our benchmark we mainly focused on three families of tasks within the Monsterkong domain, it is fairly straightforward to introduce variations of map configurations to the framework. Monsterkong provides two degrees of design choices 1. the task definitions and 2. the evolution of tasks referred to as experiment definitions. Due to the nature of how tasks are specified through simple matrices (map configurations), many layers of complexity can be added through the task specification. For example, object addition and removal can induce local variations in reward, nails can be penalizing, diamonds can be bonuses. Additionally, changes to the textures of the game like simple changes to the color of the walls, the coins, and the background as well as changes in the lighting are easy to add for users interested to test generalization of the policies learned.

### E.3 HALFCHEETAH-GRAVITY AND HOPPER-BODYPARTS

HalfCheetah-gravity and Hopper-Bodyparts are two benchmarks introduced in Mendez et al. (2020). In the first, each task consist of a different gravity. In the latter, the agent's body parts are changing in size at each tasks. The gravity and body parts values are sampled as in Mendez et al. (2020). The two benchmarks we study are each composed of 10 tasks.

# F  EXTENDED EXPERIMENTS

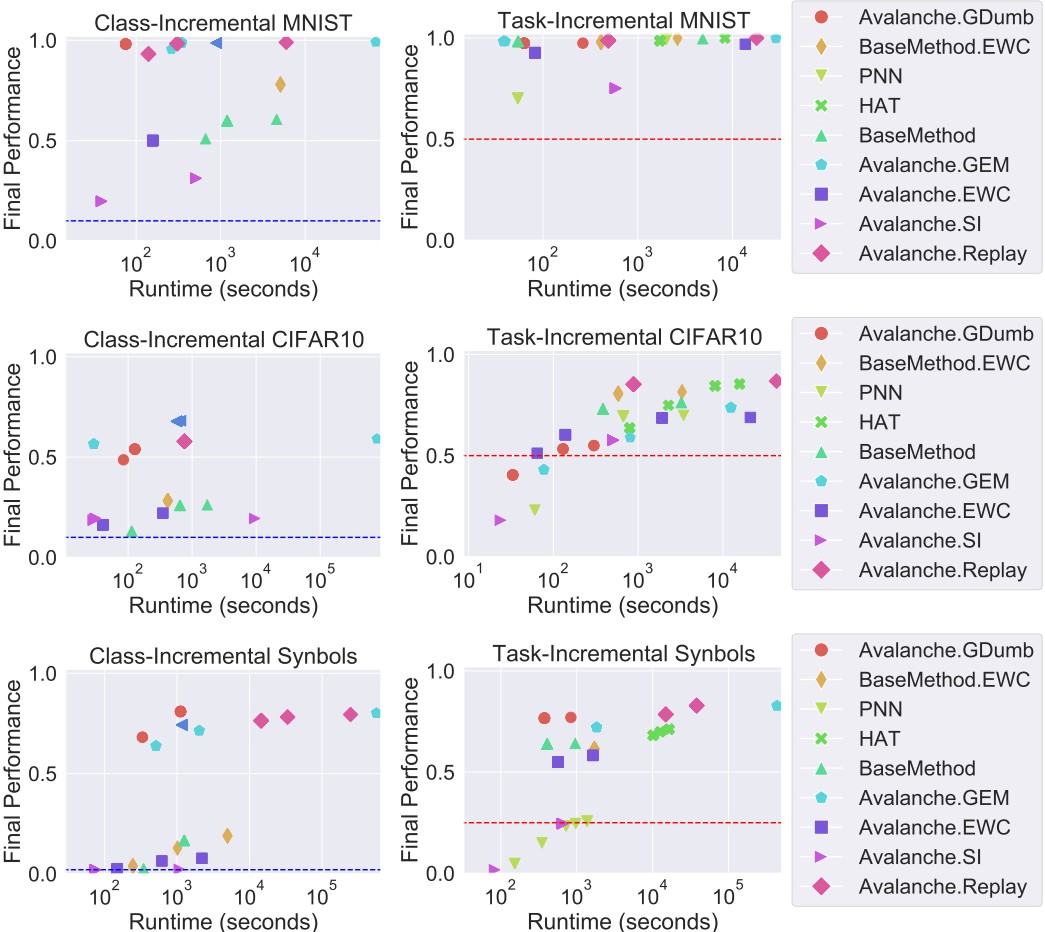

Figure 9: **Incremental Supervised Learning results**. Transpose of Figure 3 for improved readability. Final performance (vertical axis) is plotted against runtime (horizontal axis). The methods achieving the best trade-off lie closer to the top-left of the figures. The dotted line shows chance accuracy for each setting-dataset combination. For each methods, several trials are presented depending on metrics composed of linear combination of final performance and (normalized) runtime. Intuitively, better performance in CL normally comes at the cost of increased computation. This intuition is reflected in the presented results, as highlighted by the observed correlation between final performance and runtime. GEM and GDumb achieve the best tradoff, although the latter cannot make predictions in an online manner and thus serves more as a reference point.

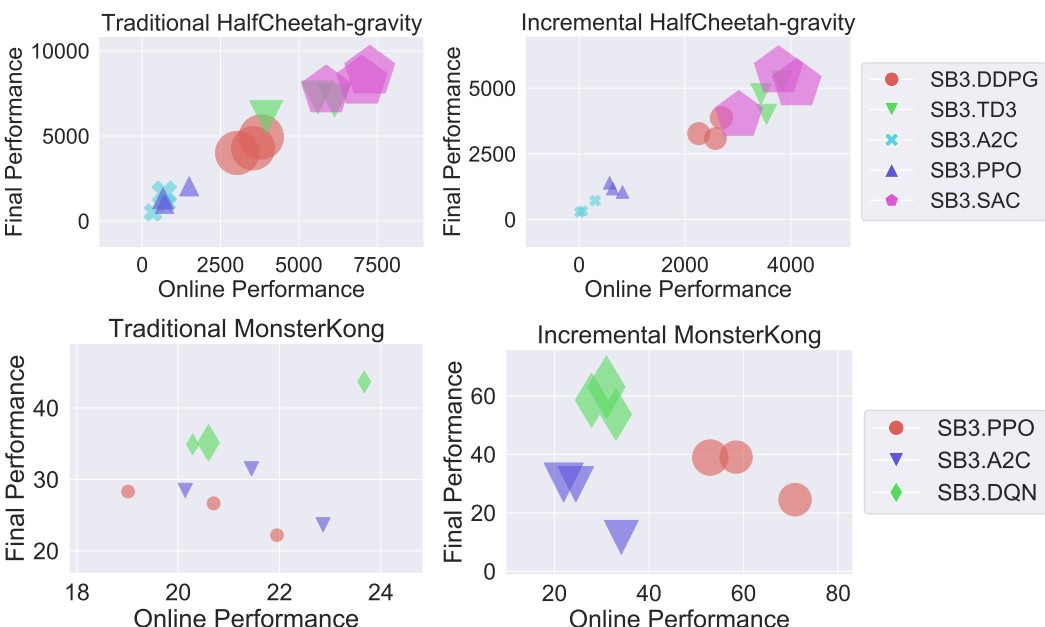

Figure 10: **Impact of the RL backbone algorithm in Traditional and Incremental RL**. Larger version of Figure 4 for improved readability. Final performance (vertical axis) is plotted against online performance (horizontal axis). The bubbles' size indicates the normalized runtime of the methods. Datasets are presented in each row and settings are presented in each column. For each method, several trials are presented depending on metrics composed of linear combination of final performance and online performance. The methods achieving the best trade-off lie closer to the top-right of the figures and have smaller bubble size. In general, we observe a trade-off between performance and runtime, a tendency also observed in Figure 9. Another interesting trade-off can be observed between final performance and online performance. E.g., in both MonsterKong benchmarks, DQN achieves the best final performance whereas PPO achieves the best online performance. Because the former is off-policy, it can re-use the previously acquired data to retain its performance on past tasks, increasing final performance. Contrarily, the latter, being on-policy, focuses on the current task and thus learns it faster, thereby increasing its online performance.

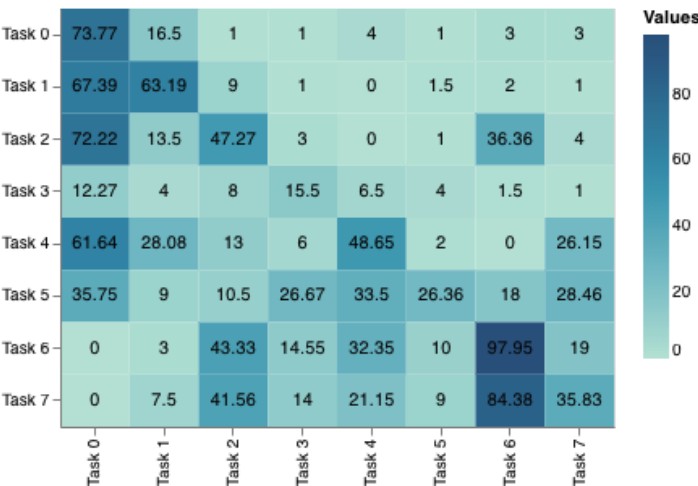

Figure 11: **PPO's Transfer matrix in Continual-MonsterKong.** Each cell at row $i$ and column $j$ indicates the test performance on task $j$ after having learned tasks 0 through $i$. The contents of each cell correspond to the average reward per episode obtained in the test environment for the corresponding task. Positive numbers above the diagonal indicate generalization to unseen tasks, which is achievable by design in the Continual-Monsterkong benchmark.

