# OpenReview forum: "Sequoia: A Software Framework to Unify Continual Learning Research"
_ICLR.cc/2022/Conference — ICLR 2022 Submitted_

### Official Review · Reviewer_YwgU · 2021-11-02

**Correctness:** 3
**Technical Novelty And Significance:** 1
**Empirical Novelty And Significance:** 2
**Recommendation:** 5
**Confidence:** 4

**Main Review:**

- In section 2, the authors propose a Markovian process that is being ignored in the later parts. It is not clear why this hidden-mode Markov decision process is useful to the proposed framework.
- While I do appreciate the effort in developing this framework, I am lacking the novelty in the paper. The work is mainly an engineering effort that's appreciated but might not fit this conference.


**Summary Of The Paper:**

This paper introduces a new continual learning framework that aims to boost the research in the field. This framework is based on a taxonomy of all possible assumptions that are common to CL methods. Moreover, this taxonomy helps in putting supervised and reinforcement methods in a unified framework.


**Summary Of The Review:**

While I do appreciate the effort behind this work, I doubt the match between this paper and the scope of ICLR.

---

> ### Author Response · Authors · 2021-11-23
> **Reply to Reviewer #4**
>
> ## HMMDP is ignored later
> The purpose of the HMMDP is to start from a general-enough mathematical framework that can recover the different flavors of CL. Our most general setting, namely Continuous Task-agnostic CL, can be explained via an HMMDP. The “Hidden-mode” part of the HMMDP disappears once the context/task variable is observed, i.e. in incremental learning.
>
> ## Submission is not a good fit for the conference
> We respectfully disagree with the reviewer on this point, as applications in vision and Software platforms are some of the topics described in the ICLR call for papers.

---

### Official Review · Reviewer_pX2k · 2021-11-02

**Correctness:** 3
**Technical Novelty And Significance:** 2
**Empirical Novelty And Significance:** 1
**Recommendation:** 3
**Confidence:** 5

**Main Review:**

The goal of the paper is very ambitious. There exist many open source libraries implementing CL methods for supervised learning and reinforcement learning settings.

FRAMEWORK:
According to the paper "each setting is described as a set of assumptions. A tree-shaped hierarchy emerges from this view". This shows a limitation of the theoretical framework. Defining settings as a set of assumptions does not result in a tree-shaped hierarchy (it would be a lattice). Therefore, using a tree-shaped hierarchy means that some methods will be compatible in principle but not in practice (in Sequoia) because the tree lacks the connection. This a strong limitation of the framework that is not discussed.

IMPLEMENTATION:
The framework heavily relies on several dependencies to implement the heavy lifting (datasets, RL environments, Methods). This is not a big problem by itself, but it is not clear what Sequoia is adding compared to the original libraries.

EXPERIMENTS:
The experiments show several baselines in the supervised and reinforcement learning settings. Most methods are not implemented by Sequoia and are inherited from avalanche for SL and stable-baselines and continual-world for RL. One thing that I expected from this section was how the same method could be easily applied to different settings, which is the main claim of the library. Instead, SL and RL settings share zero methods. For example, the EWC implementation is different in the SL and RL settings. Does this mean that users have to implement every method twice? It seems that RL and SL methods are completely separate, which is against the entire spirit of the library. At this point, what is the advantage that Sequoia brings compared to its dependencies (continuum, gym, avalanche, stable-baselines, continual-world)?

ADDITIONAL COMMENTS:
- "constraints often relate to memory, compute, or time allowed to learn a task" -> is Sequoia able to check time and memory constraints?
- is it possible for the end users to define new settings or modify the hierarchy? it would be interesting to see an example.
- "We note that some Avalanche methods achieve lower than chance accuracy in task-IL because they do not use the task label to mask out the classes that lie outside the tested task" -> can you fix this problem? Otherwise a user needs to know the internals of each library to debug the code. This adds a lot of complexity.


**Summary Of The Paper:**

The paper proposes a theoretical framework to organize research problems in the continual learning (CL) domain according to a hierarchy. This theoretical framework is used as the basic foundation for Sequoia, a software library designed to reuse methods (i.e. training algorithms) across different research problems (settings).

**Summary Of The Review:**

I believe the objective of this paper of merging together continual SL and RL is very important and ambitious. However, the paper in the present form has several weaknesses. It is unclear whether the theoretical framework really achieves the paper’s objective. Sequoia (the software) seems to be still at a very alpha stage in its development cycle. I see very little unification in the methods, which is the main scope of the paper. This strongly limits the methods' reuse between the different settings. It is unclear what advantages sequoia is bringing compared to using its dependencies directly.

---

> ### Author Response · Authors · 2021-11-23
> **Reply to Reviewer #3**
>
> We sincerely wish to thank the reviewer for his very insightful feedback, which demonstrates a very clear understanding of our intent in this work.
>
> ## Tree vs Lattice
> We thank the reviewer for this very interesting observation. The reviewer is right, our unifying framework is indeed a lattice and not a tree.  We however question whether a lattice could also be represented in code in a way that would still provide the key feature of polymorphism that we obtain thanks to our current, directed acyclic graph representation (which we confess to have wrongfully referred to as a tree in a few occurrences in this work). Nevertheless, we adjusted the manuscript to reflect the discussed nuance.
>
> ## Sequoia vs using original libraries.
> We reiterate our previous answer to this common concern.
>
> ## Lack of reuse of methods between RL and SL
> We agree with the reviewer’s sentiment on this point. However, we would like to point out that Sequoia can and does make methods such as those from Avalanche applicable in most of the CSL settings, and methods from Continual World applicable in almost all of the CRL settings. It is, of course, impossible for us to make methods from Avalanche applicable in Reinforcement Learning, or vice-versa for CW.
>
> The only methods which can directly be applied, “for free” in both RL and SL, are the ones that we implemented ourselves, such as those that are built as extensions to the BaseMethod and its BaseModel (EWC, PackNet), and PNN.
> During the next few months, we plan to work towards:
> - Improving the performance of the BaseMethod (and its variants) in RL (More specifically on the massively parallel physics environments provided by the Brax project)
> - Providing examples for multi-gpu / multi worker training on large CSL datasets with the BaseMethod.
>
> Once we achieve these two objectives, we will be in a position to design and perform a more comprehensive evaluation of the performance of methods that apply in both RL and SL.
>
> ## Time and memory constraints
> Time and memory consumption is measured by Weights and Biases (Wandb). Sequoia has no inherent mechanism for explicitly enforcing time or memory constraints. This interesting addition is however outside the current scope of our work. We consider that measuring these quantities is sufficient for now.
>
> ## Bug in methods from Avalanche
> We respectfully point the reader to our above comment regarding reuse of other libraries, as well as to the more detailed comparison provided below. In order to make the methods from Avalanche available and applicable on the different CSL settings in Sequoia, we had to adapt them and widen their applicability by adding, for instance, a simple task-inference mechanism.
> In this particular case (output masking in Incremental Learning), however, the issue lies within their original implementations in the Avalanche package, and was not easy to fix from our end without making significant modifications to their models and training logic.
>
> ## Creating a new setting
> We agree with this suggestion, and will include an example of how a new setting might be added as part of our next revision.

---

> > ### Comment · Reviewer_pX2k · 2021-11-25
> > **On rebuttal**
> >
> > Thanks for the clarification. I will give only a brief answer since I think we both made our points clear to each other.
> >
> > **Tree vs Lattice**: Of course I agree with you, there is a tradeoff between the representation of actual problem's constraints, programming language features, and usability for the end user. Still, it's important to notice the disconnect between the theory and implementation. The simplification of the lattice into a tree may be even an added value (simplicity vs completeness).
> >
> >  **Usage of other libraries**: I'm not against it, I just don't see the benefit of using them for training algorithms. If you want a method to work in all the possible settings, you probably need to adapt it heavily. Therefore, it seems to me that library reuse (at least for training algorithms) does not achieve your goal of wide applicability to all possible settings in the research tree. Reusing libraries for data loading is much easier. Training algorithms often have leaky abstractions. I would imagine that the problems you are having with avalanche are because of this issue.

---

### Official Review · Reviewer_D4MM · 2021-11-02

**Correctness:** 3
**Technical Novelty And Significance:** 1
**Empirical Novelty And Significance:** 3
**Recommendation:** 5
**Confidence:** 4

**Details Of Ethics Concerns:**

This project includes plenty of different methods, but the authors don’t include detailed information (such as the licenses) about the related open-source resources they use.

**Main Review:**

### Strengths

&nbsp;

- The continual learning research tree (Figure 2) is well organized and makes a lot of sense to me.

&nbsp;

- I think the motivation of building a unified framework for continual learning is meaningful. Many different continual learning papers evaluate their methods in different benchmark protocols. It is difficult for the following researchers to compare these related methods.

&nbsp;

- Detailed explanations and experimental results are provided in this paper. The authors also provide the results on wandb. It makes obtaining the detailed results for each setting very convenient for the following researchers.

&nbsp;

### Weaknesses

&nbsp;

- Some important continual learning baselines are not included, such as iCaRL (Rebuffi et al., 2017).

&nbsp;

- The authors only provide experiment results on small-scale datasets, such as MNIST and CIFAR. Most continual learning papers, such as iCaRL (Rebuffi et al., 2017) and BiC (Wu et al., 2019), provide the results on large-scale datasets (e.g., ImageNet-1k). As the authors trying to establish a new benchmark protocol, it is important to provide the results on large-scale datasets.

&nbsp;

- This project includes plenty of different methods, but the authors don’t include detailed information (such as the licenses) about the related open-source resources they use.

&nbsp;

- The definition of “incremental learning” in Section 2.1 is ambiguous. I think it is better to use class-incremental learning (or domain-incremental learning) directly. The reasons are as follows. (1) “Incremental learning” is often considered as another name of “continual learning”. It is weird to use it to denote a specific setting of continual learning. (2) You include class-IL and domain-IL in “incremental learning”, but you exclude task-IL. It is not reasonable.

&nbsp;

- The authors include too many details about the code implementation in the paper. I think it is better to move these parts to the appendix and include more experimental results and analyses.

**Summary Of The Paper:**

In this paper, the authors try to establish a unified framework for different continual learning settings. They also provide a Python library, which includes different related methods. Extensive experimental results are also provided.

**Summary Of The Review:**

Overall, I think this project will be a useful tool for the following continual learning researchers. I will recommend acceptance if the authors can address my concerns in the “weaknesses” part.

&nbsp;

### === Post-rebuttal Comments ===

I thought the authors aimed to establish a unified software framework that makes running continual learning experiments easy.
However, after reading the rebuttal, I think Sequoia has the following major issues and the authors failed to address them in the rebuttal:

- ***Sequoia heavily relies on the previous libraries, such as Avalanche and Continuum.*** I don't think this design is very friendly to the researchers. In my personal view, I prefer a framework that is easy to be understood and includes the most popular baselines. I think your framework should be designed for a researcher instead of a software engineer.

- ***Sequoia hasn't been evaluated on large-scale datasets (e.g., ImageNet-1k).*** If I need to use this framework, I need the framework can reproduce the results of the previous baselines correctly.

So, I don't think Sequoia is very useful to a researcher like me. According to my personal experience, I tend to reject this paper.

---

> ### Author Response · Authors · 2021-11-22
> **Reply to Reviewer #2**
>
> ## Missing baselines (e.g. iCarl)
> We disagree. iCarl already has an implementation in Avalanche. Re-implementing it in Sequoia is not in line with our objective, and only represents duplication of work.
> However, if we were to add iCarl to Sequoia, we would simply adapt their implementation by extending it and allowing it to be applied to the settings in Sequoia, as was done already with 11 other methods of Avalanche.
>
> ## Bigger datasets
> Again, we respectfully disagree. While we share the sentiment that CL methods should be evaluated on larger datasets, we do however point the reader to the clarifications above. Sequoia uses the Continuum package to create its continual supervised learning datasets.
>
> ## Method description / software licences
> We point the reviewer to our above reply to Reviewer #1 to this effect.
>
> ## “Incremental Learning” Definition is confusing
> - (Used interchangeably with CL to mean CL in general)
> - Class-Incremental and Domain-Incremental learning are under Incremental Learning, but not Task-Incremental learning. This is not reasonable.
>
> Although most CL research operates in an incremental-learning setting, there is a distinction to be made. Here, the increments refer to properly segmented tasks. However, some interesting continual problems lie outside incremental learning. Specifically, no task-boundary can be defined in such setting because the data distribution continuously shifts [1].
>
> ## Extraneous code details in the paper
> We respectfully disagree. Sequoia, a software framework, is the principal contribution of our work. We therefore judge that the amount of information related to code in the paper is appropriate.
>
> _______________
>
> [1] Task Agnostic Continual Learning Using Online Variational Bayes

---

> > ### Comment · Reviewer_D4MM · 2021-11-23
> > **Thanks for the feedback from the authors.**
> >
> > Thanks for the feedback from the authors.
> >
> > I thought the authors aimed to establish a unified software framework that makes running continual learning experiments easy.
> > However, after reading the rebuttal, I think Sequoia has the following major issues and the authors failed to address them in the rebuttal:
> >
> > - ***Sequoia heavily relies on the previous libraries, such as Avalanche and Continuum.*** I don't think this design is very friendly to the researchers. In my personal view, I prefer a framework that is easy to be understood and includes the most popular baselines. I think your framework should be designed for a researcher instead of a software engineer.
> >
> > - ***Sequoia hasn't been evaluated on large-scale datasets (e.g., ImageNet-1k).*** If I need to use this framework, I need the framework can reproduce the results of the previous baselines correctly.
> >
> > So, I don't think Sequoia is very useful to a researcher like me. According to my personal experience, I tend to reject this paper.

---

### Official Review · Reviewer_SrGM · 2021-11-03

**Correctness:** 4
**Technical Novelty And Significance:** 1
**Empirical Novelty And Significance:** 1
**Recommendation:** 1
**Confidence:** 4

**Main Review:**

Pros:

P1) Creating a software package which unifies RL and supervised learning approaches seems like a step in the right direction.

Cons:

C1) CL hierarchy - I think there are missing nodes in the hierarchy. For instance, it is important to also consider problem-incremental learning where the task is the same, but the input spaces are different (e.g. the dimensionality of the input changes, not just the input distribution).

C2) Related work: You mention that your framework doesn’t compete with others. It is not clear to me why someone else would not just use the other frameworks. A more in-depth discussion of the previous work is necessary. This way, it would be easier to distinguish your contributions are.

C3) Experiments: The text does not give us information on which methods are implemented by the authors. It gave me the impression that the evaluated methods have all been implemented by another software package.

C4) Experiments: The results are presented but not discussed.

The provided hierarchy of CL methods might contain an interesting insight, which relates RL and supervised learning approaches to CL. However, due to the lack of comparison to related work, I am not certain of this.


**Summary Of The Paper:**

The paper attempts to unify all CL research with a single formalism. Next, they present a software implementation of their framework. Finally, experiments are ran which demonstrate that Sequoia can be used to evaluate CL methods.

**Summary Of The Review:**

Overall, I don’t see a significant technological or conceptual contribution.

---

> ### Author Response · Authors · 2021-11-22
> **Reviewer #1**
>
> ## Missing nodes in the hierarchy
>
> We respectfully disagree.
> The reviewer’s observation is valid: the described problem-incremental setting is not included in the CL hierarchy, despite being a plausible setting for Continual Learning. This can be seen as an issue, from the perspective that the unifying framework, illustrated in the diagram of Figure 2, should include or cover all possible settings in CL.
>
> This isn't the case however.
> Our goal with this unifying framework is to demonstrate that these structures can be created and grown by considering new assumptions. We chose to limit ourselves to the minimal number of assumptions required in order to be able to include what we consider to be the most relevant settings in CL today.
>
> For example, domain-incremental or class-incremental supervised CL settings are also perfectly valid settings from the literature, but are not included in the unifying framework hierarchy. (Note: They are available in Sequoia, and they are created by starting from Incremental Learning and adding an assumption about whether the action space remains constant between tasks (domain-incremental) or grows between tasks (class-incremental)).
>
> ### Solution
> We will prevent further confusion by rewording all references to the hierarchy including "most, if not all settings in CL" to more accurately reflect our intentions in the next revision.
> We will also add more emphasis on our contribution being a mechanism for growing such trees, rather than the tree itself, which isn’t set in stone.
>
> ## Comparisons with Avalanche:
> We respectfully disagree with the reviewer. We point the reviewer to our above comments with respect to comparisons with other libraries, and to the more detailed description included at the end of this page. We do concede that the distinction might not be obvious at first.
>
> ### Solution:
> See above (comparisons with other libraries)
>
> ## About missing descriptions and open-source licenses
>
> We appreciate and share the reviewer’s sentiment with respect to giving proper credit to open-source libraries and licensing.
>
> ### Solution
> (see the solution to common concerns: comparisons with other frameworks): In the next revision, we will include a more detailed description of the methods from other open-source libraries in the appendix, along with references for each method and include their license information in each source file.

---

### Author Response · Authors · 2021-11-22
**Adressing Common Concerns**

We would first like to thank the reviewers for their time and very insightful feedback.

## 1. Clarifying the contribution / novelty

> "What's the point of using Sequoia? Why not use the other frameworks directly? Lack of novelty / Not enough description and comparison with related works (e.g. Avalanche)."

We thank the reviewers for bringing up this criticism, which is relevant.
We’ll use a comparison/metaphor to illustrate what I believe is at the core of this:

- https://xkcd.com/927/

I believe this comic illustrates the essence of the criticism here: why use Sequoia, if you could instead just use the frameworks directly?

What Sequoia isn’t:
- https://xkcd.com/1406/

What Sequoia is:
- https://drive.google.com/file/d/1x3DLhD3ik3kzmxZNNE_E1fQ9Gsi5bfwN/view?usp=sharing

What we're proposing isn't a "competing standard". It's a mechanism to organize the different standards into a hierarchy.
If we considered the cable type example above, then Sequoia wouldn't be a new type of universal adapter either!
It would be a structure / diagram showing what charger you can use, given the type of port on any given device.
The inheritance relationship here would denote forward-compatibility. For example, a USB type B cable can be safely plugged into a USB type A port. A Television that requires a HDMI 1.2 cable can only accept a cable of type HDMI 1.2 or HDMI 2.0, etc.

Translating this back into our domain of interest:
Sequoia is an organized catalog of settings and methods from CRL and CSL.
Users are encouraged to use whichever framework they want! We're just creating the settings in such a way that, when you create a Method and select a "target" setting from the hierarchy, your method can be applied not just on that target setting, but onto any setting at or below your target setting in the hierarchy.
We argue that the main advantage of using Sequoia is code reuse and reproducibility.
E.g. You could totally just extend the PPO algorithm from SB3 directly, rather than start from the sb3.PPOMethod in Sequoia that uses it.
The point is that later, if someone else wants to start from what you had, and apply it into a different setting, they would probably have a hard time.

## 2. Comparisons with related work (other libraries)

Sequoia builds upon and extends libraries for RL,  CRL and CSL by making their methods available to use on the various settings.

The vast majority of methods are implemented in specialized libraries (Avalanche, Stable-Baselines3, Continual_world) and are simply made available in Sequoia by:
exposing their configuration options and hyper-parameters
state which settings they can be applied to, by selecting a target setting from the hierarchy. (Note: this is equivalent to stating their assumptions about the CL problem).
adapting them slightly so that they can be trained using the environments/dataloaders created by the Settings in Sequoia.

Building upon and reusing related work correctly is something we deeply care about.
We never copy-paste anything. We import, extend, and customize methods from other libraries, only if need be. All credit is attributed to the original authors of the methods and to the framework that provides their implementations.

Users are more than welcome to use the implementations from these frameworks directly. It may in fact be simpler to do so, when there is only a very specific use-case in mind, and reusability of the solution isn’t a concern. However, by creating a new method in Sequoia, or making an existing algorithm available as a method in Sequoia, users are then able to test their method on all the applicable settings, as well as any setting that gets added in the future.

Some "in-house" methods are also available in Sequoia. One good example of which is the BaseMethod, the EWC method and PackNet. These are the only methods in Sequoia that can be applied to both RL and SL settings.

### Solution:

We will include new sections in the appendix to describe in more detail the contributions from each software framework (Avalanche, Continual World, SB3), as well as disambiguate the intended use-cases for each framework in comparison with Sequoia.
Note: we also include a detailed comparison with Avalanche at the end of this page for interested reviewers/readers.

---

> ### Author Response · Authors · 2021-11-23
> **Detailed comparison with Avalanche**
>
> ## Bonus: Detailed comparison with Avalanche
>
> ### Similarities:
> - Sequoia’s BaseMethod serves the same purpose as Avalanche's BaseStrategy.
> - Sequoia’s AuxiliaryTask serves the same purpose as Avalanche’s StrategyPlugin.
>     - Plugins in Avalanche are similar to callbacks in PyTorch Lightning.
>     - Sequoia’s AuxiliaryTask inherits from PL’s Callback class
>     - Plugins have but at least one additional hooks compared to Callbacks from PL: to modify the dataset before training.
> - Adding plugins on top of the BaseStrategy ~= Adding auxiliary tasks on top of the BaseMethod. For example:
>     - Avalanche's EWC Strategy :=  BaseStrategy + EWCPlugin
>     - Sequoia’s EWC Method := BaseMethod + EWC Auxiliary task.
>
> ### Differences:
> - Sequoia places no constraints on the design of a Method.
> - Sequoia delegates the responsibility of creating methods to other, more specialized libraries. It makes these methods applicable in a way that makes them reusable on any current or future applicable setting via polymorphism.
> - To illustrate the point: Sequoia could eventually include Avalanche scenarios as settings for CSL, if there were to be a clear hierarchical structure to their properties and assumptions. We instead opted for making Avalanche’s methods applicable onto our own settings.
> - We use Continuum as the source of data for our settings.
> - To the best of our knowledge, there isn’t a way in Avalanche to determine what scenarios a strategy can be used on.
> - The BaseMethod can be applied in CRL
>     - Avalanche for RL just came out, not clear how much overlap there is between RL and SL there. For instance, can we use the same CL “strategy” in both types of scenarios?
>     - (NOTE: this is made possible by its model (the BaseModel) using a different type of output head (PolicyHead instead of a ClassificationHead))
> - The BaseMethod uses PyTorch-Lightning for the training.
>     - Delegating the responsibility for the high-performance training of the model to PL is a very useful design decision!
>     - Sequoia isn’t responsible for running code on multiple GPUs or TPUs. PL can take care of that. → Less code duplication.
>     - All other features of PL (mixed precision, DP, DDP, TPUs,logging, etc) are either already usable with the BaseMethod, or could be easily enabled.
>
> When adapting the methods from Avalanche to be applicable on the CSL settings in Sequoia, we also made some additions to them in order to widen their applicability. For example, we added a simple form of task inference to them, so that they could be applied to the CSL settings in Sequoia that do not have task labels at test-time.
>
> ### How to transform Avalanche’s BaseStrategy into Sequoia’s BaseMethod:
> This is, roughly, how you could take Avalanche, and make it similar to Sequoia’s BaseMethod.
>
> 1. Create RL Scenarios that use gym environments as the “experience source”/dataset. (probably done in Avalanche-rl)
>     - Add a way to create non-stationarity in the RL environments.
> 2. Create structured objects for the data types that environments/datasets yield and accept.
>    (E.g. Observation / Action / Reward dataclasses, for example)
> 3. Refactor the BaseStrategy: Make it use a Trainer from PyTorch Lightning instead of doing the training itself.
>     - Make this BaseStrategy agnostic to RL or SL. Good way to go about this is to use a different type of output head for each setting, for instance.
>       Could also use a completely different BaseModel, if that’s simpler.
> 4. Refactor the BaseModel-equivalent in Avalanche (if any) to be a LightningModule.
> 5. Refactor the “Plugin” interface to extend the Callback interface from PL, adding necessary hooks and methods specific to CL.
>     - Make this new Plugin interface agnostic to RL/SL.
> 6. Refactor the CL plugins  in Avalanche so that they inherit from this new Plugin interface.
>    For the strategies that only apply in RL or only in SL, explicitly state that assumption programmatically using a property or registry of some sort.

---

### Decision · Program_Chairs · 2022-01-20

**Decision:**

Reject

**Comment:**

The manuscript introduces a taxonomy for organizing continual learning research settings and a software framework that realizes this taxonomy. Each continual learning setting is represented by as a set of shared assumptions (e.g., are task IDs observed or not) represented in a hierarchy, and the software is introduced with the hopes of unifying continual learning research.

The manuscript identifies a clear issue in the field: settings and methods for continual learning have proliferated so that there is little coherence in benchmarks, making progress difficult to judge. Reviewers generally agreed that the motivation of building software to help unify continual learning research was a positive.

However, reviewers also pointed to many concerns with the manuscript and software package (Sequoia) that comprises its main contribution. In particular, there is concern that the software is at an early stage of development and makes heavy use of existing libraries to function (e.g. Avalanche and Continuum). This makes it unclear what Sequioa offers over using its dependencies directly. As well, there is concern that multiple standard benchmark tasks and common methods are missing from the implementation — particularly for large scale experiments with, e.g. ImageNet-1k. In theory, the library allows extension and these might be implemented by others in the community. However, this would require that the original manuscript+software are strong enough to draw buy in from other researchers.
In sum, the manuscript+software does not yet offer a convincing starting point for researchers looking for a starting point to begin their continual learning research.